# Development of the core outcome set for herbal medicine treatment of lumbar disc herniation (COS-HM-LDH): A study protocol for a systematic review and a delphi survey

**Soo-Dam Kim**, **Changsop Yang**, **Mi Mi Ko**, **Kyung-Min Shin**, **Sungha Kim***,
**Myeong Soo Lee***

KM Science Research Division, Korea Institute of Oriental Medicine, Daejeon, Republic of Korea

* bozzol@kiom.re.kr (SK); mslee@kiom.re.kr (MSL)

## Abstract

### Background

Lumbar Disc Herniation (LDH) is a common global musculoskeletal disorder that causes a significant socioeconomic burden. Nonsurgical interventions are the primary treatment; however, herbal medicines (HM) are gaining traction for their efficacy in managing LDH. Conversely, their integration into clinical settings is limited owing to inconsistencies in clinical trial methodologies, mainly stemming from a lack of standardized outcome measures. Our research, highlighting this gap, sought to develop a Core Outcome Set (COS) to consistently assess HM treatments for LDH in Korean medicine (KM) clinical institutions using a consensus-driven, multi-phased approach.

### Methods

The development of the COS-HM-LDH involves four main phases: (1) a project management group (PMG) conducts a systematic review to curate an initial list of outcomes; (2) several Delphi survey rounds are administered to a selected panel of experts to validate and refine these outcomes; (3) KM clinicians are involved in a subsequent Delphi survey to further refine the outcomes; and (4) a final consensus meeting finalizes the COS.

### Discussion

The lack of a standardized COS for HM treatment of LDH in KM introduces biases and discrepancies in clinical practice. The proposed COS aims to provide a unified framework, given the challenges faced in clinical trials, especially in KM institutions. This initiative is expected to strengthen evidence quality, foster greater trust, and facilitate wider clinical implementation of HM treatment for LDH.

which permits unrestricted use, distribution, and reproduction in any medium, provided the original author and source are credited.

**Data availability statement:** No datasets were generated or analysed during the current study. All relevant data from this study will be made available upon study completion.

**Funding:** This study was supported by the Korea Institute of Oriental Medicine (grant number KSN2122211). Dr. Changsop Yang is the Principal Investigator (PI) of the funding award. The funders had no role in study design, data collection and analysis, decision to publish, or preparation of the manuscript.

**Competing interests:** The authors have declared that no competing interests exist.

**Abbreviations:** LDH, Lumbar Disc Herniation; HM, herbal medicines; COS, Core Outcome Set; KM, Korean medicine; COMET, Core Outcome Measures in Effectiveness Trials; VAS, Visual Analog Scale; COS-STAD, Core Outcome Set-STAndards for Development; EMs, effect modifiers; NSAID, Non-Steroidal Anti-Inflammatory Drug; JOA, Japanese Orthopedic Association; COS-STAR, Core Outcome Set-STAndards for Reporting; PMG, Project management group

## Trial registration

The Core Outcome Measures in Effectiveness Trials (COMET) Initiative: https://www.comet-initiative.org/Studies/Details/2909

## Background

Lumbar disc herniation (LDH), a prevalent medical condition globally, involves displacing the spinal disc's inner gelatinous core, known as the nucleus pulposus, through its protective fibrous ring, predominantly in the posterolateral region [1]. Approximately 95% of disc herniations in the lumbar area occur at L4-L5 or L5-S1 [2,3]. Notably, such protrusions often compress adjacent nerve roots, causing typical clinical manifestations, such as low back pain, sciatica, muscle weakness, sensory deficits, and signs of nerve root tension, with symptoms that often radiate down the legs [3–6]. LDH affects millions globally, with an estimated 1–3% of the population experiencing symptoms [7]. This condition is most prevalent among those aged 30–50 years, and men are twice as likely to be affected as women [6,7]. According to Jung et al. (2020), South Korea reported 478,697 cases in 2016, approximately 918.2 patients per 100,000 individuals [8]. The socioeconomic implications of LDH extend beyond the health of individuals and represent a significant burden on healthcare systems and economies [9]. Direct medical costs, lost workdays and reduced productivity highlight LDH's financial and societal toll [10,11].

Nonsurgical treatments for LDH, such as physical therapy, pain management using Non-Steroidal Anti-Inflammatory Drugs (NSAIDs), and lifestyle modifications, are typically employed as primary first-line treatments for LDH. Surgical interventions are considered when conservative treatments fail [12–14]. Herbal medicine (HM) is increasingly considered an alternative to conservative treatment. Notably, numerous studies have suggested that herbal therapies can relieve pain, reduce inflammation, and enhance the overall well-being of patients with LDH [15–19]. According to meta-analyses, patients treated with HM exhibited significant improvements in the Japanese Orthopedic Association (JOA) scores and decreased scores on the Visual Analog Scale (VAS) compared with a control group [20,21]. Furthermore, another meta-analysis highlighted the superior clinical efficacy of Shentong Zhuyu Decoction over conventional treatments in managing LDH, showing significant improvements in pain scores and a reduced incidence of adverse events [22].

However, although herbal treatments for LDH show potential, their broader acceptance and integration into clinical settings are hindered by methodological inconsistencies in clinical trials [19–22]. Consequently, this challenge stems primarily from a need for standardized measurements [23]. Notably, different studies have used diverse endpoints, methodologies, and assessment scales, resulting in a fragmented understanding of the efficacy of herbal treatments. These disparities introduce data inconsistency and potential reporting bias, leading to patient irrelevance and a lack of consensus on a standardized outcome [24]. Therefore, such inconsistencies complicate the decision-making process of healthcare practitioners and raise uncertainties regarding patients exploring alternative or complementary treatment options.

To address these concerns, a recent nationwide web-based survey of 500 LDH patients in South Korea investigated their real-world symptoms and priorities [25]. The most common symptoms reported were leg numbness (87.0%) and back pain (81.2%), while the most common disabilities were discomfort in sitting (64.6%) and lifting (63.6%). Back pain (48.4%) was identified as the highest priority symptom for improvement. Importantly, a majority of patients expressed preference for improvement in disability over pain (55.8%), a stable effect over a rapid effect (78.2%), and safety over treatment efficacy (56.4%). Safety (25.8%) and cost (22.2%) were also recognized as important treatment factors. Based on these findings, we aim to identify the essential outcome measures that should be prioritized in LDH management.

A core outcome set (COS) represents a consensus-driven selection of crucial outcome measures that stakeholders believe should be consistently reported in clinical trials for a given medical field [24,26]. The Core Outcome Measures in Effectiveness Trials (COMET) Initiative emphasizes the importance of standardized and consistent trial reporting to ensure robust research comparisons [26]. Using a systematic and consensus-driven approach, our research seeks to identify the essential outcomes and bolster the credibility of HM for LDH management. This effort addresses the urgent need for a standardized outcome model that can be used in both clinical settings and research. Our goal is to develop consistent evaluation criteria to assess the effectiveness of herbal treatments for LDH in both Korean Medicine (KM) clinical institutions and clinical trials.

## Methods

In developing the COS-HM-LDH, we adhere to the recommendations of the Core Outcome Set-STAndards for Development (COS-STAD) and the Core Outcome Set-STAndards for Reporting (COS-STAR) [27,28]. The methodology includes four strategic phases: (1) establishment of a project management group (PMG) that undertakes a systematic review, from which an initial list of outcomes is curated and subsequently refined; (2) conducting a series of Delphi survey rounds with a chosen panel of experts to ensure the validity and relevance of the outcomes; (3) engaging KM clinicians in another Delphi survey to further refine the identified outcomes; and (4) undertaking a consensus meeting to finalize the COS. The structure and phases of this study are shown in Fig 1. This research started in September 2023 and is registered in the COMET database under the registration No. 2909, which can be accessed at https://www.comet-initiative.org/Studies/Details/2909. The recruitment of participants for the Delphi survey began in July 2024.

### Phase 1: project management group and systematic review

**Project management group.** We establish a PMG comprising six researchers from the Korea Institute of Oriental Medicine to oversee and conduct a comprehensive systematic review, extract outcomes/effect modifiers (EMs), and eliminate duplicate findings. The PMG includes researchers, physicians, statisticians, and methodologists. Having statisticians in the PMG ensures rigorous data analysis and interpretation.

**Systematic review.** A systematic review comprehensively explores and catalogs the outcomes pertinent to LDH in line with the recommendations of the COMET initiative [29]. Notably, studies published in Korean (OASIS, Science-On) and English (PubMed, CENTRAL) databases are considered for all study types. The detailed search strategy for the English databases can be found in the S1 Table.

We focus on patients diagnosed with LDH who are treated with HM, either as a standalone treatment or in combination with other treatments. Inclusion is based on clear LDH diagnostic criteria, and all patients are considered, irrespective of sex, age, or ethnicity. Studies are excluded if they primarily investigate outcomes of complications associated with LDH, such as diseases caused by LDH, or involve patients with spinal stenosis or other severe spinal conditions. If a study focused more on secondary LDH conditions than on primary LDH, its complete text will be scrutinized to evaluate its eligibility for inclusion.

Two independent researchers evaluate the titles and abstracts to determine the relevance of each study. Using Microsoft Excel, data such as author(s), publication year, patient demographics, study type, intervention, comparator, outcomes,

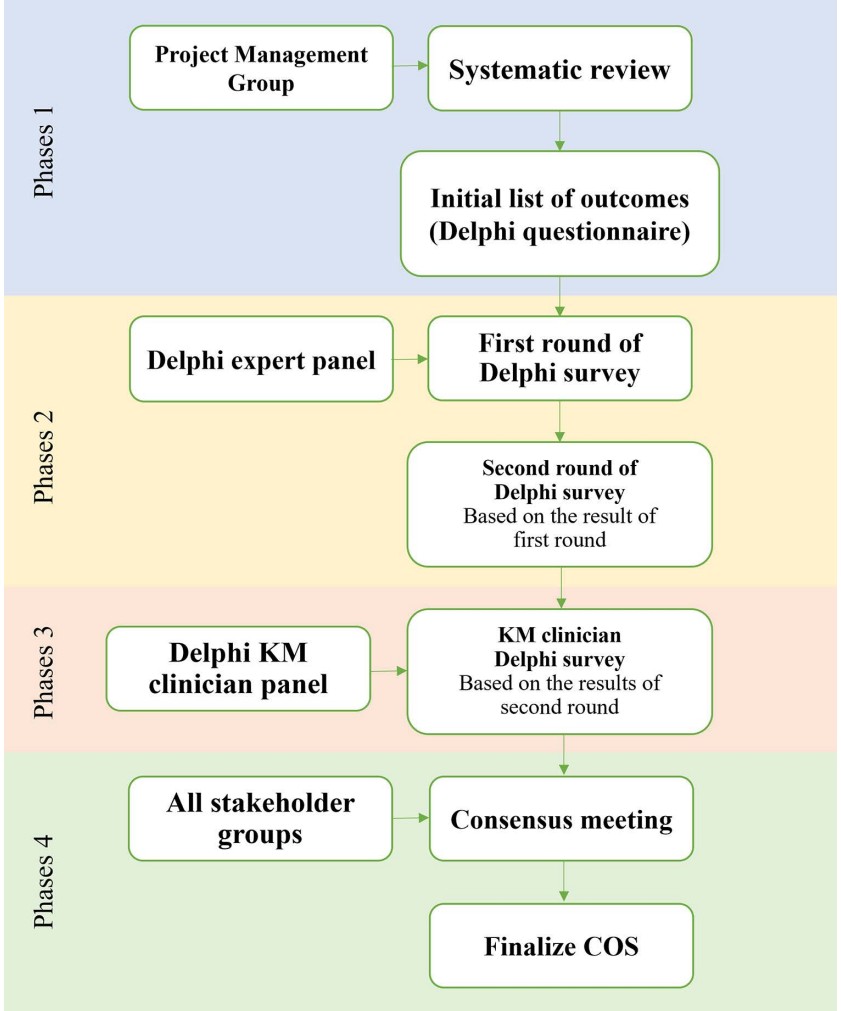

**Fig 1. Flow chart of study design.** KM: Korean medicine; COS: core outcome set.

treatment duration, follow-up length, and outcome measurement methods are extracted from each study. Any extraction discrepancies or study evaluation disagreements are resolved through discussion or consultation with a third expert. Duplicate studies and those not providing comprehensive LDH outcomes are excluded. Similar outcomes are recorded only once and thoughtfully combined and condensed to maintain the content's integrity.

In addition, previously published books and articles, such as "Korean medicine clinical practice guideline for lumbar herniated intervertebral disc" and other relevant sources [30,31], are reviewed to extract the recommended outcomes and EMs for LDH.

**Development of initial list of COS.** After reviewing the collected outcomes and EMs, the PMG refines and finalizes a list consistent with the objectives of this COS. All outcomes extracted from the systematic review are included in the initial list to ensure comprehensiveness. For each outcome, the number of studies in which it was reported will also be presented, thereby providing information on the frequency and strength of use in previous research. In addition, to reflect patients' perspectives, findings from the previously conducted nationwide survey of LDH patients are incorporated, highlighting patient-prioritized symptoms and treatment needs [25]. These combined sources form the basis of the initial COS questionnaire, which will be used for the first Delphi process.

## Phase 2: expert panel and delphi survey

The Delphi method, an iterative consensus technique, gathers expert opinions and streamlines the outcomes into a prioritized list. It comprises successive surveys conducted anonymously by a diverse panel, ensuring that each participant's perspective is vital [32].

**Expert panel.** There are no specific guidelines for determining the number of participants in a Delphi panel [33]. However, it is generally accepted that larger sample sizes in Delphi studies enhance reliability [34]. Therefore, we aim to include as many panelists as feasible, drawing on the sample sizes from previous COS studies [35–37]. We convene a comprehensive panel of stakeholders, targeting an expected number of 15–30 participants. This panel includes researchers from various fields (such as KM experts, statistics, nursing, and policy experts) and specialized clinicians directly involved in the care and management of patients in the region where the COS is being developed (Table 1).

The selection of participants is based on recommendations from professional societies associated with LDH, such as The Korean Acupuncture & Moxibustion Medicine Society and the Society of Rehabilitation Medicine of Korean Medicine. Additionally, the identified experts are encouraged to recommend other individuals they consider suitable for inclusion in this initiative. The criteria for selection follow the COS-STAD guidelines, which provide a set of 11 minimum standards for the development of core outcome sets [28].

**First round Delphi.** We employ a structured Delphi survey to achieve a consensus on pertinent outcomes and EMs. The initial list of outcomes is methodically grouped by topic and sorted alphabetically within each group to minimize bias. Using a 9-point Likert scale [38], where 1–3 signifies an outcome of "limited importance," 4–6 "important but not critical," and 7–9 "critical," recommended by the COMET Initiative [39], panelists rate items based on their significance. During the initial round, participants evaluate the importance of pre-identified outcomes while suggesting any additional outcomes they deem vital. Any outcomes suggested as new are incorporated into the second round of Delphi. Notably, all scored outcomes from the first round are also included in the second round of Delphi. Subsequent rounds consider feedback from earlier stages to ensure a comprehensive review and discussion of all outcomes.

In the case of patients or their representatives, feedback is substituted with results from previous survey research that examined the actual symptoms and needs of lumbar disc herniation patients [25]. During the expert Delphi process, the results of this research are presented, ensuring that the perspectives of patients are adequately considered.

**Second round Delphi.** Panelists who participated in the first round of Delphi are invited to participate in the second round. At this stage, they are presented with Questionnaire 2, in which they rescore the previously evaluated outcomes. We provide them with the scores assigned during the first round for informed decision-making. In addition, a detailed descriptive statistical analysis from the first round is provided, encompassing each outcome's responses and score distribution. This approach ensures that the panelists can reevaluate and adjust their scores by considering the perspectives and insights of their respective stakeholder groups. We plan to convene a panel of KM clinicians based on

**Table 1. Overview of stakeholder participation.**

| Key stakeholders | Expected Number | Allocation (%) | Selection Criteria |
|---|---|---|---|
| Researchers (various fields including KM experts, statistics, nursing, policy experts) | 10-20 | 67% | Conducted at least one clinical study related to lumbar disc herniation in the past five years, or be registered in the clinical trials registry system to conduct related studies within the next two years. |
| Specialized Clinicians (directly involved in the care and management of patients in the COS development region) | 5-10 | 33% | Must have at least 5 years of clinical experience related to lumbar disc herniation and handle more than 30 patients annually. |
| Patients or Their Representatives | – | – | Replaced by the survey results from previous research [25]. |

COS: core outcome set; KM: Korean medicine

the outcomes of the second round of Delphi. This panel aims to review and assess the clinical relevance and validity of the identified COS, ensuring that our research is robust and directly applicable to clinical settings.

## Phase 3: KM clinician panel and Delphi survey

We plan to implement this unique step to ensure the feasibility and relevance of the identified outcomes in a real-world clinical setting. KM clinicians chosen to represent key stakeholder perspectives review the proposed outcomes and use a Delphi survey. At the PMG request, the Association of Korean Medicine (https://akom.org/English/Index) recommends KM clinician panels with at least 5 years of clinical experience in LDH. It is anticipated that these clinicians possess expertise in treating and managing LDH. The Delphi survey of these KM clinicians focuses on reviewing and evaluating the derived outcomes concerning their feasibility in KM clinical institutions. Notably, all evaluations are performed using a 9-point scale to ascertain the relevance and applicability of each outcome.

Following the completion of the third round of the Delphi survey, the identified outcomes are categorized into one of three domains recommended by the COMET initiative [39]: "consensus out," "consensus in," or "without consensus," based on the criteria provided in Table 2. Adopting these predetermined definitions reduces the risk of post-hoc consensus determination and limits biases from the preconceived notions of the research team [40]. Subsequently, the findings from this phase are forwarded to a consensus meeting for further deliberation and finalization.

## Phase 4: consensus meeting

To obtain a holistic perspective on LDH, we ensure that representatives from all stakeholder groups participate in the consensus meeting. Those who diligently complete the Delphi survey are invited. The consensus group comprises the PMG, Delphi expert panel, and a KM clinician panel.

A consensus meeting is convened at a dedicated location to finalize the COS for LDH. The discussions are informed by the results collated from each Delphi round, focusing on the outcomes of the last Delphi survey. Outcomes that reached a "consensus in" status are subjected to the anonymous vote of either "yes" or "no." Outcomes receiving approval from at least 70% of the participants are integrated into the final COS for LDH. Conversely, outcomes designated as "consensus out" are excluded.

A thorough discussion follows for outcomes marked as "without consensus" in the third round. Participants then anonymously rescore these outcomes during the meeting using the familiar 9-point Likert scale. Outcomes achieving scores of 1–3 from at least 70% of the participants, with a maximum of 15% of the participants giving them scores of 7–9, are discarded. Conversely, outcomes rated 7–9 by a minimum of 70% of participants and receiving 1–3 scores by ≤15% are included in the final LDH COS. This repetitive rescoring process is repeated until a consensus on all outcomes is reached.

## Ethics approval

Ethical approval to conduct this study has been granted by The Institutional Review Board of the Korea Institute of Oriental Medicine (IRB approval no. I-2312/012-003-04). All individuals participating in the Delphi rounds will provide informed consent prior to their involvement in the study.

**Table 2. Definition of consensus.**

| Category | Description | Criteria |
| --- | --- | --- |
| Consensus out | The outcome is agreed upon to be excluded from the core outcome set | At least 70% rated 1–3, and <15% rated 7–9 across stakeholders |
| Consensus in | The outcome is agreed upon to be included in the core outcome set | <15% rated 1–3, and at least 70% rated 7–9 across stakeholders |
| Without consensus | There is no clear agreement on the inclusion/exclusion of the outcome | Other conditions |

## Discussion

The need for a standardized COS for HM treatment of LDH has posed significant challenges in clinical settings, where consistent outcomes are vital. Since more than 80% of KM institutions operate as primary care clinics, the emphasis on standardizing outcomes has become even more urgent [41]. The absence of a consistent COS can lead to potential reporting biases and discrepancies, emphasizing the need for outcomes that fit various treatment strategies [30]. However, conducting clinical trials in these settings presents challenges, such as human resource shortages, equipment limitations, and policy restrictions [42]. Introducing a COS tailored to herbal treatments for LDH in KM would provide a robust framework for future clinical trials that ensures consistent and relevant outcomes, enhancing the clarity and reliability of their findings. This progress can significantly improve the quality of evidence, building more confidence in herbal treatments for LDH and making it easier for broader use in KM institutions.

While previous studies have explored various outcomes in LDH treatment using HM, there remains no consensus on which outcomes should be prioritized or standardized [43–45]. Several reviews have reported heterogeneity in outcome selection, making it difficult to compare or synthesize findings across studies [20,46]. To date, no COS has been proposed specifically for HM in the context of LDH. This study addresses that gap by systematically developing a COS in accordance with established guidelines [27,28], aiming to improve the consistency, comparability, and relevance of future clinical research in KM. By grounding the COS in both expert consensus and real-world clinical settings, this protocol contributes to bridging the gap between research and clinical practice.

## Trial status

The COS-HM-LDH is presently at the stage of systematic review in its development process.

## Supporting information

**S1 Table. Search strategy for English databases.**
(DOC)

**S1 File. PRISMA-P-checklist.**
(DOCX)

## Author contributions

**Conceptualization:** Soo-Dam Kim, Sungha Kim.

**Data curation:** Mi Mi Ko, Kyung-Min Shin.

**Formal analysis:** Mi Mi Ko, Kyung-Min Shin.

**Methodology:** Soo-Dam Kim, Sungha Kim.

**Project administration:** Changsop Yang.

**Supervision:** Myeong Soo Lee, Sungha Kim.

**Writing – original draft:** Soo-Dam Kim.

**Writing – review & editing:** Changsop Yang, Mi Mi Ko, Kyung-Min Shin, Myeong Soo Lee, Sungha Kim.

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
