## [Decision Letter · Decision Letter 0]

21 Mar 2025

Dear Dr. Kim,

Thank you for submitting your manuscript to PLOS ONE. After careful consideration, we feel that it has merit but does not fully meet PLOS ONE’s publication criteria as it currently stands. Therefore, we invite you to submit a revised version of the manuscript that addresses the points raised during the review process.

**EDITORIAL COMMENTS**

**Editor’s comments:**

In this study, the authors intend to develop a Core Outcome Set for Herbal Medicine Treatment of Lumbar Disc Herniation (COS-HM-LDH). However, the methodology section is entirely in the future tense, indicating that the proposed steps have yet to be established and standardized.This raises questions about the study's foundation; it seems to focus on potential future actions rather than presenting a concrete protocol.To enhance clarity, the authors should address this concern directly, providing a rationale for their approach and the relevance of discussing future methodologies.An expanded discussion on the purpose of the study and its significance in the context of existing literature would also strengthen their argument.This would help clarify the study's objectives and the importance of developing standardized outcomes for herbal treatments in this area.

**Decision – Accepted with major revision**

**Please find enclosed Reviewer’s comments:**

**Reviewer 1- Accept**

Well constructed manuscript.

Looking forward for your valuable contribution to the scientific community.

**Reviewer 2 - Accept**

Well-articulated, and a good initiative to establish standard guidelines.

This study can serve as a guiding document for any further works based on complementary and alternative medicine, globally.

**Reviewer 3 – Minor Revision**

Table 1 Patients or Their Representatives why this data is originally not collected ?, graphical abstract is requird

Kind regards,

Richa Gupta

Academic Editor

PLOS ONE

Additional Editor Comments:

EDITORIAL COMMENTS

EDITORIAL COMMENTS

Editor’s comments:

In this study, the authors intend to develop a Core Outcome Set for Herbal Medicine Treatment of Lumbar Disc Herniation (COS-HM-LDH). However, the methodology section is entirely in the future tense, indicating that the proposed steps have yet to be established and standardized.

This raises questions about the study's foundation; it seems to focus on potential future actions rather than presenting a concrete protocol.

To enhance clarity, the authors should address this concern directly, providing a rationale for their approach and the relevance of discussing future methodologies.

An expanded discussion on the purpose of the study and its significance in the context of existing literature would also strengthen their argument.

This would help clarify the study's objectives and the importance of developing standardized outcomes for herbal treatments in this area.

Decision – Accepted with major revision

Please find enclosed Reviewer’s comments:

Reviewer 1- Accept

Well constructed manuscript.

Looking forward for your valuable contribution to the scientific community.

Reviewer 2 - Accept

Well-articulated, and a good initiative to establish standard guidelines.

This study can serve as a guiding document for any further works based on complementary and alternative medicine, globally.

Reviewer 3 – Minor Revision

Table 1 Patients or Their Representatives why this data is originally not collected ?, graphical abstract is requird

Reviewers' comments:

Reviewer's Responses to Questions

**Comments to the Author**

1. Does the manuscript provide a valid rationale for the proposed study, with clearly identified and justified research questions?

Reviewer #1: Yes

Reviewer #2: Yes

Reviewer #3: Yes

2. Is the protocol technically sound and planned in a manner that will lead to a meaningful outcome and allow testing the stated hypotheses?

Reviewer #1: Yes

Reviewer #2: Yes

Reviewer #3: Yes

3. Is the methodology feasible and described in sufficient detail to allow the work to be replicable?

Reviewer #1: Yes

Reviewer #2: Yes

Reviewer #3: Yes

4. Have the authors described where all data underlying the findings will be made available when the study is complete?

Reviewer #1: Yes

Reviewer #2: Yes

Reviewer #3: Yes

5. Is the manuscript presented in an intelligible fashion and written in standard English?

Reviewer #1: Yes

Reviewer #2: Yes

Reviewer #3: Yes

You may also provide optional suggestions and comments to authors that they might find helpful in planning their study.

Reviewer #1: Well constructed manuscript. Looking forward for your valuable contribution to the scientific community.

Reviewer #2: Well-articulated, and a good initiative to establish standard guidelines. This study can serve as a guiding document for any further works based on complementary and alternative medicine, globally.

Reviewer #3: Table 1 Patients or Their

Representatives why this data is originally not collected ?, graphical abstract is requird

**Do you want your identity to be public for this peer review?** For information about this choice, including consent withdrawal, please see our Privacy Policy

Reviewer #1: No

Reviewer #2: No

Reviewer #3: **Yes: ** Dr. Saurabh Singh Professor , Ayurvedic Pharmacy , School of Pharmaceutical Sciences, Lovely Professional University, Punjab

---

## [Author Response · Author response to Decision Letter 1]

18 Apr 2025

Editor’s Comments

o In this study, the authors intend to develop a Core Outcome Set for Herbal Medicine Treatment of Lumbar Disc Herniation (COS-HM-LDH). However, the methodology section is entirely in the future tense, indicating that the proposed steps have yet to be established and standardized.

o This raises questions about the study's foundation; it seems to focus on potential future actions rather than presenting a concrete protocol.

o To enhance clarity, the authors should address this concern directly, providing a rationale for their approach and the relevance of discussing future methodologies.

o An expanded discussion on the purpose of the study and its significance in the context of existing literature would also strengthen their argument.

o This would help clarify the study's objectives and the importance of developing standardized outcomes for herbal treatments in this area.

Response:

- Thank you for your thoughtful comments. In response, we have revised the methodology section by changing the tense from future to present to clearly reflect the protocol nature of the study and to distinguish between ongoing and completed components. Additionally, we have expanded the discussion section to better articulate the significance of this study in the context of existing literature, emphasizing the current lack of standardized outcome measures for herbal medicine treatment of LDH and how this study addresses that gap. Revised parts have been marked in red in the manuscript.

Reviewer 1 - Accept

Well constructed manuscript. Looking forward for your valuable contribution to the scientific community.

Response:

- We sincerely thank the reviewer for their positive and encouraging feedback.

Reviewer 2 - Accept

Well-articulated, and a good initiative to establish standard guidelines. This study can serve as a guiding document for any further works based on complementary and alternative medicine, globally.

Response:

- We are grateful for the reviewer’s kind words and appreciation of our efforts. We hope this study will contribute meaningfully to future standardization in herbal medicine research.

Reviewer 3 – Minor Revision

Table 1 Patients or Their Representatives – why this data is originally not collected?, Graphical abstract is required.

Response:

- We appreciate the reviewer’s valuable suggestions. In this study, we aimed to reflect patient perspectives by incorporating data from a previously published nationwide survey that investigated the real-world symptoms and needs of patients with LDH (Kim et al., 2023), rather than recruiting patients or their representatives directly. This approach enabled us to ensure the inclusion of patient-centered outcomes while avoiding duplication of efforts and addressing feasibility constraints within the current research scope. To further enhance the clarity and accessibility of the study, we have also created and included a graphical abstract that visually summarizes the background, aim, methodological phases, and expected impact of the COS-HM-LDH development.

---

## [Editor Report · Decision Letter 1]

21 May 2025

PONE-D-24-26777R1

Development of the Core Outcome Set for Herbal Medicine Treatment of Lumbar Disc Herniation (COS-HM-LDH): A Study Protocol for a Systematic Review and a Delphi Survey

PLOS ONE

Dear Dr. Kim,

Thank you for submitting your manuscript for review by PLOS ONE. After careful consideration, we have decided that your manuscript does not meet our standards for systematic and scoping reviews. We are therefore rejecting your manuscript.

PLOS ONE considers protocols for systematic reviews, scoping reviews and meta-analyses for publication; however, these submissions must address a clear and defined research question; outline a systematic and comprehensive literature review; and describe clearly reported, reproducible, and systematic methods to identify, select, and critically appraise relevant research. Our author guidelines for systematic reviews are at https://journals.plos.org/plosone/s/submission-guidelines#loc-systematic-reviews-and-meta-analyses.

In this case, we note that there is no planned assessment of the quality or risk of bias of the included studies. Conclusions in a systematic review or meta-analysis should be related to the quality of the included publications. Without this critical assessment of the included studies, we do not feel that your systematic review currently fulfills our requirements for consideration (for more information, we would refer you to the following source: Boutron I, Page MJ, Higgins JPT, Altman DG, Lundh A, Hróbjartsson A. Chapter 7: Considering bias and conflicts of interest among the included studies [last updated August 2022]. In: Higgins JPT, Thomas J, Chandler J, Cumpston M, Li T, Page MJ, Welch VA (editors). Cochrane Handbook for Systematic Reviews of Interventions version 6.5. Cochrane, 2024. Available from www.training.cochrane.org/handbook).

Additionally, there is no assessment of the certainty of the body of evidence for each outcome planned. This is distinct from assessments of the quality or risk of bias of the included studies, and is essential to inform further healthcare decisions based on the results of the review. We do not feel that valid conclusions can be drawn from your review without this assessment  (for more information, please see Schünemann HJ, Higgins JPT, Vist GE, Glasziou P, Akl EA, Skoetz N, Guyatt GH. Chapter 14: Completing ‘Summary of findings’ tables and grading the certainty of the evidence [last updated August 2023]. In: Higgins JPT, Thomas J, Chandler J, Cumpston M, Li T, Page MJ, Welch VA (editors). Cochrane Handbook for Systematic Reviews of Interventions version 6.5. Cochrane, 2024. Available from www.training.cochrane.org/handbook). 

As a result of these concerns we are unable to consider your manuscript. I am sorry that we cannot be more positive on this occasion, but I hope you understand the reasons for this decision.

Kind regards,

Emma Campbell, Ph.D

Staff Editor

PLOS One

- - - - -

---

## [Author Response · Author response to Decision Letter 2]

14 Jul 2025

Editor’s Comments

o In this study, the authors intend to develop a Core Outcome Set for Herbal Medicine Treatment of Lumbar Disc Herniation (COS-HM-LDH). However, the methodology section is entirely in the future tense, indicating that the proposed steps have yet to be established and standardized.

o This raises questions about the study's foundation; it seems to focus on potential future actions rather than presenting a concrete protocol.

o To enhance clarity, the authors should address this concern directly, providing a rationale for their approach and the relevance of discussing future methodologies.

o An expanded discussion on the purpose of the study and its significance in the context of existing literature would also strengthen their argument.

o This would help clarify the study's objectives and the importance of developing standardized outcomes for herbal treatments in this area.

Response:

- Thank you for your thoughtful comments. In response, we have revised the methodology section by changing the tense from future to present to clearly reflect the protocol nature of the study and to distinguish between ongoing and completed components. Additionally, we have expanded the discussion section to better articulate the significance of this study in the context of existing literature, emphasizing the current lack of standardized outcome measures for herbal medicine treatment of LDH and how this study addresses that gap. Revised parts have been marked in red in the manuscript.

Reviewer 1 - Accept

Well constructed manuscript. Looking forward for your valuable contribution to the scientific community.

Response:

- We sincerely thank the reviewer for their positive and encouraging feedback.

Reviewer 2 - Accept

Well-articulated, and a good initiative to establish standard guidelines. This study can serve as a guiding document for any further works based on complementary and alternative medicine, globally.

Response:

- We are grateful for the reviewer’s kind words and appreciation of our efforts. We hope this study will contribute meaningfully to future standardization in herbal medicine research.

Reviewer 3 – Minor Revision

Table 1 Patients or Their Representatives – why this data is originally not collected?, Graphical abstract is required.

Response:

- We appreciate the reviewer’s valuable suggestions. In this study, we aimed to reflect patient perspectives by incorporating data from a previously published nationwide survey that investigated the real-world symptoms and needs of patients with LDH (Kim et al., 2023), rather than recruiting patients or their representatives directly. This approach enabled us to ensure the inclusion of patient-centered outcomes while avoiding duplication of efforts and addressing feasibility constraints within the current research scope. To further enhance the clarity and accessibility of the study, we have also created and included a graphical abstract that visually summarizes the background, aim, methodological phases, and expected impact of the COS-HM-LDH development.

---

## [Decision Letter · Decision Letter 2]

11 Sep 2025

Dear Dr. Kim,

Thank you for submitting your manuscript to PLOS ONE. After careful consideration, we feel that it has merit but does not fully meet PLOS ONE’s publication criteria as it currently stands. Therefore, we invite you to submit a revised version of the manuscript that addresses the points raised during the review process.

We look forward to receiving your revised manuscript.

Kind regards,

Richa Gupta

Academic Editor

PLOS ONE

Journal Requirements:

1. Your ethics statement should only appear in the Methods section of your manuscript. If your ethics statement is written in any section besides the Methods, please delete it from any other section.

Additional Editor Comments (if provided):

This is an interesting and timely manuscript in which the authors aim to develop a Core Outcome Set for Herbal Medicine Treatment of Lumbar Disc Herniation (COS-HM-LDH). The idea is indeed innovative and has the potential to establish much-needed uniformity in outcome reporting, thereby opening new avenues for research and clinical practice in this area. If successfully conceptualized and translated into practical application, such a standardized set could greatly enhance the comparability of future studies, reduce reporting bias, and ultimately improve evidence synthesis for clinical decision-making.

However, after a careful and critical review, there remain several important issues that need to be adequately addressed before the manuscript can be considered ready for publication. These issues not only pertain to methodological clarity and robustness but also to the practical feasibility of implementing the proposed Core Outcome Set in real-world clinical settings. Moreover, reviewers have raised certain specific concerns and questions that warrant a more detailed response from the authors. Addressing these points comprehensively will strengthen the credibility, reproducibility, and eventual impact of this important work.

Decision – Accepted after major changes

Please find attached reviewer’s comments below

Thank you so much for the opportunity to review your manuscript.

Background: offers clear rationale to the study, highlighting the need for establishing a COS to evaluate the effectiveness of herbal medicine in lumbar disc herniation management.

Methodology – The four-phase development process (systematic review, Delphi surveys with experts and clinicians, and a consensus meeting) is well organized and follows established COS development frameworks. The consensus method is described here a priori. The figure offers visual summary of the proposed protocol.

Development of the Initial COS List: this step is not very clear.

What is the criteria that will be used to synthesis the initial set of COS? Will the PMG employ theme saturation or another qualitative research method?

The patients’ perspective should be considered in this step, as per standard 8 of COS-STAD.

The authors note that patient participation has been replaced by reference to a previously published patient survey on the same topic. Including a brief summary of this survey in the introduction would enhance the manuscript’s value. Furthermore, can the survey findings be objectively integrated into the Delphi process outcomes? If not, the inclusion of patient representatives in the Delphi rounds perhaps should be considered?

I commend the authors for their thorough and well-structured approach and hope to see the results of their effort published out soon.

Reviewers' comments:

Reviewer's Responses to Questions

**Comments to the Author**

1. Does the manuscript provide a valid rationale for the proposed study, with clearly identified and justified research questions?

Reviewer #4: Yes

2. Is the protocol technically sound and planned in a manner that will lead to a meaningful outcome and allow testing the stated hypotheses?

Reviewer #4: Yes

3. Is the methodology feasible and described in sufficient detail to allow the work to be replicable?

Reviewer #4: Yes

4. Have the authors described where all data underlying the findings will be made available when the study is complete?

Reviewer #4: Yes

5. Is the manuscript presented in an intelligible fashion and written in standard English?

Reviewer #4: Yes

You may also provide optional suggestions and comments to authors that they might find helpful in planning their study.

Reviewer #4: Thank you so much for the opportunity to review your manuscript.

Background: offers clear rationale to the study, highlighting the need for establishing a COS to evaluate the effectiveness of herbal medicine in lumbar disc herniation management.

Methodology – The four-phase development process (systematic review, Delphi surveys with experts and clinicians, and a consensus meeting) is well organized and follows established COS development frameworks. The consensus method is described here a priori. The figure offers visual summary of the proposed protocol.

- Development of the Initial COS List: this step is not very clear.

What is the criteria that will be used to synthesis the initial set of COS? Will the PMG employ theme saturation or another qualitative research method?

The patients’ perspective should be considered in this step, as per standard 8 of COS-STAD.

The authors note that patient participation has been replaced by reference to a previously published patient survey on the same topic. Including a brief summary of this survey in the introduction would enhance the manuscript’s value. Furthermore, can the survey findings be objectively integrated into the Delphi process outcomes? If not, the inclusion of patient representatives in the Delphi rounds perhaps should be considered?

I commend the authors for their thorough and well-structured approach and hope to see the results of their effort published out soon.

**Do you want your identity to be public for this peer review?** For information about this choice, including consent withdrawal, please see our Privacy Policy

Reviewer #4: No

---

## [Author Response · Author response to Decision Letter 3]

22 Sep 2025

Journal Requirements

Comment: Your ethics statement should only appear in the Methods section of your manuscript. If your ethics statement is written in any section besides the Methods, please delete it from any other section.

Response: Thank you for this important reminder. We have removed the “Ethics declarations” section from the end of the manuscript and relocated the corresponding content to the Methods section under a new subsection titled “Ethical approval.” This revision ensures that the ethics statement appears only in the Methods section, in line with the journal’s requirements.

Revised Sentence (Page 13):

“Ethical approval to conduct this study has been granted by The Institutional Review Board of the Korea Institute of Oriental Medicine (IRB approval no. I-2312/012-003-04). All individuals participating in the Delphi rounds will provide informed consent prior to their involvement in the study.”

Reviewer #4

Comment 1: Background: offers clear rationale to the study, highlighting the need for establishing a COS to evaluate the effectiveness of herbal medicine in lumbar disc herniation management.

Response: We thank the reviewer for this positive feedback.

Comment 2: Methodology – The four-phase development process (systematic review, Delphi surveys with experts and clinicians, and a consensus meeting) is well organized and follows established COS development frameworks. The consensus method is described here a priori. The figure offers visual summary of the proposed protocol.

Response: We thank the reviewer for this positive and encouraging feedback on our methodological framework.

Comment 3: Development of the Initial COS List: this step is not very clear. What is the criteria that will be used to synthesis the initial set of COS? Will the PMG employ theme saturation or another qualitative research method? The patients’ perspective should be considered in this step, as per standard 8 of COS-STAD.

Response: Thank you for this important comment. We have clarified that the PMG will include all outcomes from the systematic review, merge similar items, and report their frequency across studies to form the initial COS list. To meet COS-STAD standard 8, we also incorporated findings from a nationwide patient survey, ensuring that patient-prioritized symptoms and treatment needs are reflected in the initial questionnaire for the Delphi process.

Revised Sentence (Page 7-8):

“After reviewing the collected outcomes and EMs, the PMG refines and finalizes a list consistent with the objectives of this COS. All outcomes extracted from the systematic review are included in the initial list to ensure comprehensiveness. For each outcome, the number of studies in which it was reported will also be presented, thereby providing information on the frequency and strength of use in previous research. In addition, to reflect patients’ perspectives, findings from the previously conducted nationwide survey of LDH patients are incorporated, highlighting patient-prioritized symptoms and treatment needs [25]. These combined sources form the basis of the initial COS questionnaire, which will be used for the first Delphi process.”

Comment 4: The authors note that patient participation has been replaced by reference to a previously published patient survey on the same topic. Including a brief summary of this survey in the introduction would enhance the manuscript’s value.

Response: As suggested, we have added a concise summary of the nationwide patient survey in the Background section, highlighting priority symptoms (e.g., back pain, leg pain, functional disability) and treatment preferences (e.g., safety and stable effect).

Revised Sentence (Page 4-5):

“To address these concerns, a recent nationwide web-based survey of 500 LDH patients in South Korea investigated their real-world symptoms and priorities [25]. The most common symptoms reported were leg numbness (87.0%) and back pain (81.2%), while the most common disabilities were discomfort in sitting (64.6%) and lifting (63.6%). Back pain (48.4%) was identified as the highest priority symptom for improvement. Importantly, a majority of patients expressed preference for improvement in disability over pain (55.8%), a stable effect over a rapid effect (78.2%), and safety over treatment efficacy (56.4%). Safety (25.8%) and cost (22.2%) were also recognized as important treatment factors. Based on these findings, we aim to identify the essential outcome measures that should be prioritized in LDH management.”

Comment 5: Furthermore, can the survey findings be objectively integrated into the Delphi process outcomes? If not, the inclusion of patient representatives in the Delphi rounds perhaps should be considered?

Response: Thank you for this helpful comment. We revised the Methods section to state that the findings from the nationwide patient survey have been incorporated into the development of the initial COS list. These patient-prioritized outcomes form part of the first Delphi questionnaire, ensuring that patient perspectives are objectively reflected in the Delphi process.

Revised Sentence (Page 7-8):

“…In addition, to reflect patients’ perspectives, findings from the previously conducted nationwide survey of LDH patients are incorporated, highlighting patient-prioritized symptoms and treatment needs [25]. These combined sources form the basis of the initial COS questionnaire, which will be used for the first Delphi process.”

Comment 6: I commend the authors for their thorough and well-structured approach and hope to see the results of their effort published out soon.

Response: We sincerely thank the reviewer for the encouraging comments.

We are encouraged by this feedback and will strive to ensure that the study produces meaningful results.

---

## [Decision Letter · Decision Letter 3]

15 Oct 2025

Development of the Core Outcome Set for Herbal Medicine Treatment of Lumbar Disc Herniation (COS-HM-LDH): A Study Protocol for a Systematic Review and a Delphi Survey

PONE-D-24-26777R3

Dear Dr. Kim,

We’re pleased to inform you that your manuscript has been judged scientifically suitable for publication and will be formally accepted for publication once it meets all outstanding technical requirements.

Kind regards,

Richa Gupta

Academic Editor

PLOS ONE

Additional Editor Comments (optional):

I have carefully reviewed the revised version of the manuscript entitled “Development of the Core Outcome Set for Herbal Medicine Treatment of Lumbar Disc Herniation (COS-HM-LDH): A Study Protocol for a Systematic Review and a Delphi Survey” (Manuscript ID: [PONE-D-24-26777R3] which was resubmitted following the reviewers comments. I am pleased to note that the authors have made extensive and meaningful revisions in accordance with all the suggestions provided during the review process. I appreciate the authors’ diligent efforts in addressing all points raised during the review and commend them for significantly improving the quality of their work.

Upon detailed examination, I find that the authors have:

• Addressed all critical and minor comments comprehensively and thoughtfully.

• Enhanced the clarity and scientific accuracy of the manuscript through careful reorganization and precise language refinement.

• Strengthened the methodological description, ensuring reproducibility and transparency.

• Updated and contextualized their findings within the current scientific literature.

• Provided appropriate justifications where requested, thereby improving the robustness of the conclusions.

The revised manuscript now reflects a high standard of scientific quality and coherence. The presentation of results is clear, the discussion is balanced and well-substantiated, and the conclusions are consistent with the data presented. Importantly, the study contributes novel and relevant insights to the existing body of knowledge in the field of Herbal Medicine Treatment of Lumbar Disc Herniation, making it suitable for publication.

In light of these observations, I am fully satisfied with the authors’ responses and the quality of the revised submission. Therefore, I recommend

Acceptance of the manuscript

Reviewers' comments:

Reviewer's Responses to Questions

**Comments to the Author**

1. Does the manuscript provide a valid rationale for the proposed study, with clearly identified and justified research questions?

Reviewer #4: Yes

2. Is the protocol technically sound and planned in a manner that will lead to a meaningful outcome and allow testing the stated hypotheses?

Reviewer #4: Yes

3. Is the methodology feasible and described in sufficient detail to allow the work to be replicable?

Reviewer #4: Yes

4. Have the authors described where all data underlying the findings will be made available when the study is complete?

Reviewer #4: Yes

5. Is the manuscript presented in an intelligible fashion and written in standard English?

Reviewer #4: Yes

You may also provide optional suggestions and comments to authors that they might find helpful in planning their study.

Reviewer #4: Thank you for taking my comments into consideration. I hope you find them useful to enhance the quality of your great work.

**Do you want your identity to be public for this peer review?** For information about this choice, including consent withdrawal, please see our Privacy Policy

Reviewer #4: No

---

## [Editor Report · Acceptance letter]

PONE-D-24-26777R3

PLOS ONE

Dear Dr. Kim,

I'm pleased to inform you that your manuscript has been deemed suitable for publication in PLOS ONE. Congratulations! Your manuscript is now being handed over to our production team.

Kind regards,

on behalf of

Dr. Richa Gupta

Academic Editor

PLOS ONE